# A New Clear-Sky Method for Assessing Photosynthetically Active Radiation at the Surface Level

**William Wandji Nyamsi [1],\*, Philippe Blanc [2], John A. Augustine [3]** , **Antti Arola [1] and Lucien Wald [2]**

[1]   Finnish Meteorological Institute, 70211 Kuopio, Finland; antti.arola@fmi.fi
[2]   MINES ParisTech, PSL Research University, Centre Observation, Impacts, Energy, 06904 Sophia Antipolis, France; philippe.blanc@mines-paristech.fr (P.B.); lucien.wald@mines-paristech.fr (L.W.)
[3]   NOAA Earth System Research Laboratory, Global Monitoring Division (GMD), Boulder, CO 80305, USA; john.a.augustine@noaa.gov
\*   Correspondence: william.wandji@fmi.fi; Tel.: +358-50-304-8221

**Abstract:** A clear–sky method to estimate the photosynthetically active radiation (PAR) at the surface level in cloudless atmospheres is presented and validated. It uses a fast and accurate approximation adopted in several radiative transfer models, known as the *k*-distribution method and the correlated-*k* approximation, which gives a set of fluxes accumulated over 32 established wavelength intervals. A resampling technique, followed by a summation, are applied over the wavelength range [0.4, 0.7] μm in order to retrieve the PAR fluxes. The method uses as inputs the total column contents of ozone and water vapor, and optical properties of aerosols provided by the Copernicus Atmosphere Monitoring Service. To validate the method, its outcomes were compared to instantaneous global photosynthetic photon flux density (PPFD) measurements acquired at seven experimental sites of the Surface Radiation Budget Network (SURFRAD) located in various climates in the USA. The bias lies in the interval $[-12, 61]$ μmol m$^{-2}$ s$^{-1}$ ($[-1, 5]$ % in values relative to the means of the measurements at each station). The root mean square error ranges between 37 μmol m$^{-2}$ s$^{-1}$ (3%) and 82 μmol m$^{-2}$ s$^{-1}$ (6%). The squared correlation coefficient fluctuates from 0.97 to 0.99. This comparison demonstrates the high level of accuracy of the presented method, which offers an accurate estimate of PAR fluxes in cloudless atmospheres at high spatial and temporal resolutions useful for several bio geophysical models.

**Keywords:** photosynthetically active radiation; correlated–*k* approximation; resampling technique; Copernicus Atmosphere Monitoring Service; cloudless atmospheres; albedo

## 1. Introduction

Photosynthetically active radiation (PAR) is the part of solar radiation which lies in the wavelength range of [0.4, 0.7] μm. PAR plays a key role in the biomass production and more precisely in the growth of plants through the photosynthesis process [1–3]. PAR is the incident power per unit surface area; its unit is W m$^{-2}$. PAR is also a measure of the amount of photons per time unit per surface unit, called the photosynthetic photon flux density (PPFD), whose unit is μmol m$^{-2}$ s$^{-1}$. The widely used approximation of McCree (1972) [4] relates the PAR and the PPFD: 1 W m$^{-2}$ ≈ 4.57 μmol m$^{-2}$ s$^{-1}$.

Researchers and other specialists in ecophysiological, agricultural, and bio-geophysical domains demand high quality estimates of PAR and of its direct and diffuse components. Both components summed together give the global PAR. The diffuse and direct components have diverse effects on the

plants. For instance, Li et al. [5] have reported that diffuse illumination produces a more homogeneous illumination profile in the canopy than direct illumination.

Suitable instruments, such as quantum sensors ([6]) are often used to meet the increasing needs in PAR measurements. Because of instrument costs, maintenance and operation, PAR measurements are still sparse over time and in space. Researchers have looked at proxies for PAR to overcome this paucity, especially at the broadband or the total solar radiation because measurements of the latter are more often available in the world [7–9]. Several authors proposed a constant proportion between the daily means of broadband irradiance and PAR. Udo and Aro [10] suggested a proportionality coefficient equal to 2.079 µmol J$^{-1}$ while Jacovides et al. [11] suggested 1.919. The clear benefit of using such an approach is the availability of accurate estimates of broadband irradiance derived from satellite images (see e.g., [12–15]). Alternate sources for broadband irradiance are meteorological analyses, though they offer lower quality ([16–18]).

Researchers recognize that the proportionality coefficient should be a function of atmospheric variables. To that extent, several methods have been developed for estimating PAR in cloudless atmospheres by using different sources of atmospheric measurements as predictor variables. In all-sky conditions, the effects of clouds are accounted for separately by an appropriate attenuation of the cloudless PAR, also called the cloud modification factor [19]. Su et al. [20] have proposed such a method with inputs on atmospheric conditions from Clouds and the Earth's Radiant Energy System (CERES) products. Their method mostly shows a positive relative bias reaching up to 7% when validated with PAR measurements made under cloudless atmospheres at seven experimental sites of the Surface Radiation Budget Network (SURFRAD) in the USA [21]. Following the same principle, Bosch et al. [22] have developed a parametrization for PAR in cloud-free conditions and found a relative bias of less than 1% when validated at three SURFRAD sites. More recently, Sun et al. [23] have proposed a method using solar radiation in the Ultraviolet (UV)–visible spectral band. They found mostly a negative bias of less than 4% at the seven SURFRAD sites when using ground-based atmospheric measurements as inputs. We want to emphasize that in these two latter methods, the necessary inputs were obtained from ground-based measurements from specific selected stations, while in Su et al. [20], they were based on satellite measurements. Thus, their performance cannot be directly compared. The methods utilizing satellite measurements as inputs (e.g., [20]) can provide a global coverage, which is a clear advantage. The method using ground-based measurements can provide estimates only at specific locations. However, the advantage of using ground-based measurements as inputs is that their accuracy is better, which translates directly into a better accuracy of the PAR estimates. Therefore, it is important to make a distinction between these two types of methods, when comparing the performance of these PAR estimation methods in general.

Atmospheric radiative transfer models are usually computationally expensive but are the best way to obtain accurate PAR estimates, provided an accurate description of the cloudless atmospheres and ground properties are put into the model. The library for radiative transfer (libRadtran) [24,25] is such a model and was used in this work. The *k*-distribution method and correlated-*k* approximation of Kato et al. [26] represents one scheme adopted by libRadtran as well as the Doubling Adding KNMI (Koninklijk Nederlands Meteorologisch Instituut) Royal Netherlands Meteorological Institute (DAK), Rapid Radiative (RAPRAD) transfer, and SPECMAGIC models to speed-up computations in order to produce broadband irradiance. In this scheme, only 32 spectral bands over the large spectral range of [0.240, 4.606] µm are used to calculate the broadband irradiance. The 32 results are summed up to yield the broadband irradiance. From now on, these 32 spectral intervals are called Kato bands (KB), with the band number in subscript. Wandji Nyamsi et al. [27] compared the transmissivities calculated by the Kato et al. [26] scheme for each of the 32 KBs to those resulting from detailed spectral calculations. They concluded that estimates from the Kato et al. scheme are accurate and useful for representing irradiances in each of the eleven KBs covering the PAR band. Besides its proven accuracy, this scheme may be used to set up operational chains. One example is the operational McClear clear-sky model

that provides the total irradiance in cloudless atmospheres by making use of several abaci, also known as look-up tables that have been pre-computed with libRadtran ([28,29]).

The eleven KBs covering the PAR spectral interval are KB #6 [0.363, 0.408] µm to KB #16 [0.684, 0.704] µm. Wandji Nyamsi et al. [30] have reported on a resampling technique over the PAR range that gives the irradiance for every 1 nm within each KB over the PAR range from the irradiance of 30 nm width of each KB in any atmospheric state in cloudless conditions. The technique is explained in more detail in Section 3.2. PAR estimates from the technique have been compared against PAR that has been simulated from detailed spectral calculations of libRadtran serving as a reference. The comparison has revealed a very high accuracy, much better than that of the methods based on a proportionality coefficient proposed by Udo and Aro [10] and by Jacovides et al. [11].

This resampling technique has not been validated in operational conditions for assessing PAR fluxes, i.e., using inputs on atmospheric conditions and ground properties from satellite observations or from meteorological numerical models. The objective of this paper is to make this step by describing the entire method and evaluating its results against the measured PAR in cloudless conditions. We note that while our ultimate goal is to produce an all-sky algorithm, we advance in a step-by-step approach, and it is crucial to separately assess the performance of the clear-sky algorithm, to be better able to later understand the performance of the modelling of the cloud effects. The measurements of the PAR fluxes were collected at the seven stations of the SURFRAD network. The atmospheric properties, namely, the total column contents of ozone (TOC) and water vapor (TWV), and the optical properties of aerosols, were collected from the Copernicus Atmosphere Monitoring Service (CAMS). The ground reflective properties are from a series of maps, proposed by Blanc et al. [31], of the Moderate Resolution Imaging Spectroradiometer (MODIS)-derived Bi-directional Reflectance Distribution Function (BRDF) parameters for each calendar month. This article describes the first step of a wider project aiming at generating a ready-to-use method for estimating PAR in all-sky conditions by taking advantage of the real time availability of the CAMS products. Practically, the new method will contribute noticeably in the near future by offering accurate estimates of PAR fluxes at a high spatial and temporal resolution, thus providing an essential variable to bio-geophysical models.

## 2. Ground–Based Measurements Used

Figure 1 displays the location of the seven SURFRAD stations used, while Table 1 reports their geographical coordinates and identifying codes. The global PPFDs are measured at the SURFRAD sites with the LI–COR Quantum sensor model LI–190. The direct and diffuse components of the PAR are not measured. These high-quality measurements were downloaded at SURFRAD FTP Server [32]. Seven full years of 1 min averages of the PAR were collected, from 01-01-2010 to 31-12-2016. In addition, the broadband diffuse and global irradiances in the broadband interval [0.28, 2.8] µm, direct broadband irradiances at a normal incidence, UV-B fluxes, thermal infrared fluxes, air temperature, relative humidity, station pressure and wind speed, measured every 1 min, were also downloaded.

The accuracy of all of the quantum sensors is ±5% [21]. The manufacturer of the LI-COR instrument reports a total error of approximately 8% ([6]). A possible calibration drift of the Quantum sensor is checked by replacing each sensor every year with freshly calibrated units, and by visually monitoring the possible degradation of the PAR measurement, as recommended by the Baseline Surface Radiation Network [33]. This quality control consists of computing each day the ratio of the daily mean of the PAR to the daily mean of the broadband irradiance. This ratio may lie in the interval [0.4, 0.65], depending on the sun elevation, the TOC and TWV, and the optical properties of aerosols and clouds [34]. If this ratio reaches below 0.4 and continues to decrease over several days, the instrument is replaced and a correction is applied on the PAR data from the instant when the drift occurred.

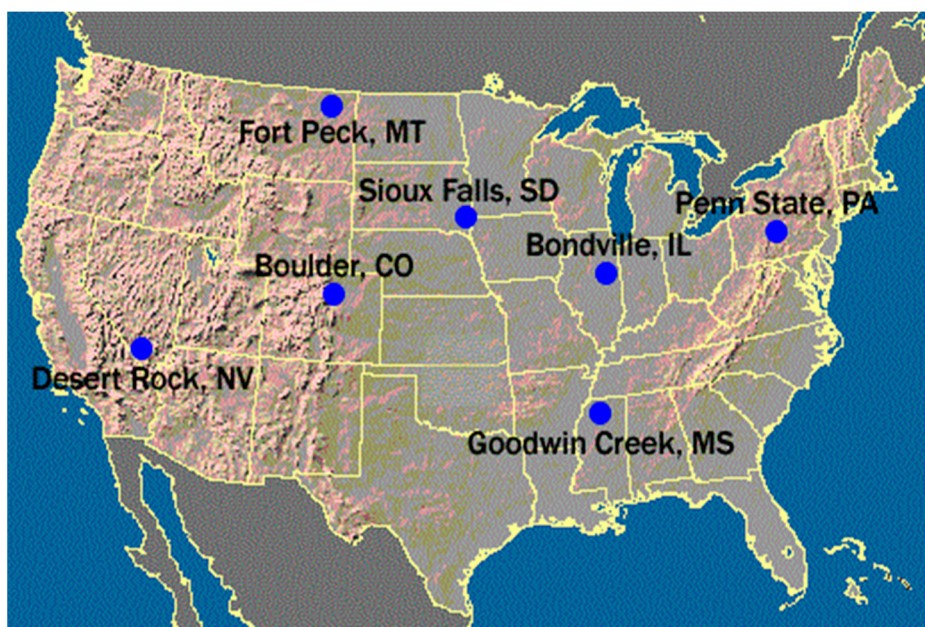

**Figure 1.** Map of the seven SURFRAD sites (courtesy of the NOAA).

**Table 1.** Description of the ground-based stations used for the measurement collection, from the northernmost station to the southernmost one.

| Station | Fort Peck | Sioux Falls | Penn. Sate Univ | Table Mountain | Bondville | Desert Rock | Goodwin Creek |
|---|---|---|---|---|---|---|---|
| Code | FPK | SXF | PSU | TBL | BND | DRA | GCM |
| Latitude (°) | 48.31 | 43.73 | 40.72 | 40.12 | 40.05 | 36.62 | 34.25 |
| Longitude (°) | −105.10 | −96.62 | −77.93 | −105.24 | −88.37 | −116.02 | −89.87 |
| Elevation amsl * (m) | 634 | 473 | 376 | 1689 | 213 | 1007 | 98 |
| NCLP ** | 186,698 | 245,355 | 120,097 | 260,509 | 196,871 | 603,727 | 230,420 |

* amsl: above mean sea level. ** NCLP: Number of cloudless periods.

The stations experience different climates and different ground properties. The Fort Peck station lies in a flat agricultural area with grasses and few trees, like the Bondville one. Fort Peck experiences a high interannual variation in snow cover. The Sioux Falls station is situated on herbaceous grounds near the Earth Resources Observation and Science (EROS) Data Center. The Penn. State Univ. station lies on an agricultural research farm in a wide Appalachian valley and is surrounded by grass with crop in the southwest quarter. Similarly, the Goodwin Creek station is situated in a rural pasture. The surface surrounding the Table Mountain station is sandy with a mix of exposed rocks, small cacti, desert shrubs, and sparse grasses. The flora is usually green in the late spring and early summer, and browns significantly by midsummer. The Desert Rock station is also in a desert-type landscape and experiences a hot arid climate; the surroundings are mostly made up of small rocks and desert shrubs, with no noticeable seasonal change in the vegetation.

We assumed that the cloudless periods identified by analyzing the broadband irradiances are also cloudless periods in the PAR measurements. Hence, cloudless periods can be more accurately detected by using the broadband direct, diffuse and global irradiances, i.e., three measurements, instead of a single measurement of the global PAR. The algorithm of Lefèvre et al. [15] has been applied to the time series of the broadband direct, diffuse and global irradiances at each site to yield a series of detected cloudless periods. We recognize that in certain cases, the PAR may be affected by scattered cloudiness which may go unnoticed in the broadband measurements and that it is possible that the retained series of cloudless periods may include cloudy periods for the PAR. Because the PAR is a major contributor to

the broadband irradiance, and given the high selectivity of the Lefèvre et al. [15] algorithm, we believe that such cases are rare and that the conclusions will be unaffected as a whole. Two consecutive filters compose the algorithm. The first filter only retains those values for which the ratio of the diffuse to the global irradiance is under 0.3. The second filter computes the ratio of the global broadband irradiance to the product of the broadband irradiance received at the top of the atmosphere and a typical air mass, and inspects the temporal variability of this quantity, which should be steady for several hours in cloudless conditions. The number of cloudless 1-min periods (NCLP) that were detected at each site is given in Table 1.

## 3. Method

In brief, the method computes the transmissivities in the 11 KBs covering the PAR range plus KB#17 ([0.704, 0.743] µm), through the Kato et al. [26] scheme, before performing a spectral resampling of the transmissivities every 1 nm, converting in fluxes and aggregating the resampled fluxes in the interval [0.4, 0.7] µm.

### 3.1. Inputs to Libradtran

In cloudless atmospheres, the PAR depends mostly on the solar zenith angle $\theta_s$, the TOC and TWV, the aerosol optical depth (AOD) and type, the vertical profiles of the temperature, pressure, density, and volume mixing ratio for gases, the elevation of the site above the mean sea level, and the ground albedo. The origins of the inputs are selected, taking into account that the method will be used in an operational mode to provide estimates of the PAR–irradiance and PPFD–at any location and any time. $\theta_s$ is given by the SG2 algorithm [35]. The TOC and TWV, and the aerosol optical depths for organic matter, black carbon, dust, sea salt, and sulfate, originate from CAMS. The five vertical profiles are: tropics (coded afglt), mid-latitude summer and winter (afglmls and afglmlw), and sub-Arctic summer and winter (afglss and afglsw), from the Air Force Geophysics Laboratory (AFGL) data sets. A map of weights by Gschwind et al. [36] indicates which ones to use at any location, as well as their respective weights. The digital terrain model is the so-called SRTM data set that derives from the Shuttle Radar Topography Mission. When no information on the type of surface and on the ground albedo in the PAR is available, we have adopted the approach of Bosch et al. [22], where the PAR albedo is equal to 0.47 times the broadband albedo. The albedo is defined as the ratio of the global upwelling irradiance to the global downwelling irradiance in a given spectral band. It is also defined as the integral of the bidirectional reflectance distribution function (BRDF), depending on the surface-type and its roughness. Here, the broadband albedo is given by the series of maps of Blanc et al. [31] that provide the MODIS-derived BRDF parameters for each calendar month with no missing values at a spatial resolution of 0.05°.

For simplicity and convenience, the retrieval of inputs is performed by machine-to-machine requests to the McClear web service on the Soda website (Gschwind et al. [28], www.soda-pro.com, last access: 14-04-2019). The flow returned by the service in the verbose mode contains 1 min values of the inputs listed above that are conveniently exploited for the validation.

### 3.2. Description of the Spectral Resampling and the Proposed Method

The resampling technique has been presented in Wandji Nyamsi et al. [30,37] for the PAR range and UV range, respectively. The technique is a pure modelling concept with radiative transfer simulations with libRadtran. No measurements have been used for its development.

First, a set of 60,000 atmospheric condition parameters in cloudless atmospheres has been built with Monte-Carlo draws, following the statistical distribution of each input, as reported in Table 2 in Wandji Nyamsi et al. [30]. For each condition, libRadtran is run twice for both the direct and global irradiances: one with the Kato et al. [26] scheme and the other with the detailed spectral calculations every 1 nm. Then, the irradiances are converted into transmissivities in order to eliminate

the influence resulting from the daily and annual variations of $\theta_s$, as well as from the dependency on the extraterrestrial solar spectrum.

**Table 2.** The spectral intervals i.e. Kato bands (KBs) of the Kato et al. [26] scheme and sub-intervals i.e. narrow bands (NB$_i$) used for the resampling technique. The intercepts and slopes of the affine functions to infer the transmissivity over NB$_i$ from the transmissivity over KB.

| KB | Interval Δλ, μm | Sub-Interval NB$_i$, μm (#*i*) | Direct | | Global | |
|----|----|----|----|----|----|----|
| | | | Slope | Intercept | Slope | Intercept |
| 6 | 0.363–0.408 | 0.385–0.386 (#*1*) | 0.9987 | −0.0023 | 1.0030 | −0.0032 |
| 7 | 0.408–0.452 | 0.430–0.431 (#*2*) | 1.0026 | −0.0004 | 0.9995 | 0.0013 |
| 8 | 0.452–0.518 | 0.484–0.485 (#*3*) | 1.0034 | 0.0005 | 0.9979 | 0.0000 |
| 9 | 0.518–0.540 | 0.528–0.529 (#*4*) | 0.9998 | −0.0005 | 1.0008 | −0.0013 |
| 10 | 0.540–0.550 | 0.545–0.546 (#*5*) | 1.0001 | 0.0003 | 1.0003 | −0.0003 |
| 11 | 0.550–0.567 | 0.558–0.559 (#*6*) | 1.0004 | 0.0004 | 0.9997 | 0.0012 |
| 12 | 0.567–0.605 | 0.569–0.570 (#*7*) | 0.9960 | −0.0119 | 1.0024 | −0.0100 |
| | | 0.586–0.587 (#*8*) | 1.0123 | 0.0064 | 0.9929 | 0.0267 |
| | | 0.589–0.590 (#*9*) | 0.9568 | −0.0109 | 0.9804 | −0.0434 |
| | | 0.602–0.603 (#*10*) | 1.0150 | 0.0167 | 1.0051 | 0.0212 |
| 13 | 0.605–0.625 | 0.615–0.616 (#*11*) | 1.0004 | 0.0009 | 0.9977 | 0.0033 |
| 14 | 0.625–0.667 | 0.625–0.626 (#*12*) | 1.0104 | −0.0174 | 1.0622 | −0.0551 |
| | | 0.644–0.645 (#*13*) | 1.0072 | 0.0029 | 0.9960 | 0.0154 |
| | | 0.656–0.657 (#*14*) | 0.9915 | 0.0068 | 0.9698 | 0.0205 |
| 15 | 0.667–0.684 | 0.675–0.676 (#*15*) | 1.0006 | 0.0007 | 0.9978 | 0.0036 |
| 16 | 0.684–0.704 | 0.685–0.686 (#*16*) | 1.0473 | 0.0212 | 0.9681 | 0.1036 |
| | | 0.687–0.688 (#*17*) | 0.9602 | −0.0130 | 1.0041 | −0.0531 |
| | | 0.694–0.695 (#*18*) | 0.9828 | −0.0153 | 1.0323 | −0.0642 |
| 17 | 0.704–0.743 | 0.715–0.716 (#*19*) | 1.0262 | 0.0121 | 0.9771 | 0.0596 |

For each KB$_j$ and for 1-nm spectral intervals i.e. narrow bands denoted NB$_i$, within the KB$_j$, scatterplots were made between the transmissivities of NB$_i$ and those of KB$_j$. Over the PAR range, a visual inspection of each scatterplot clearly shows a straight line with a squared correlation coefficient greater than 0.999 in all cases. Therefore, affine functions were established between the transmissivities by a least-square fitting technique. There is a considerable number of affine functions, approximately 300 for the PAR range. For operational purposes, a limited set of NB$_i$ was selected and then used in a linear interpolation process to obtain the transmissivities in all NB$_i$ without losing accuracy to compute the PAR fluxes after aggregation. A set of 19 NB$_i$ was found to be sufficient. Table 2 reports the NB$_i$ as well as the slopes and intercepts of the affine functions. The operational method is as follows. A run by libRadtran provides the fluxes in the 12 KBs. Then, the fluxes are obtained at each of the 19 NB$_i$ by using the affine functions in Table 2. Then, a linear interpolation technique is applied to these 19 known fluxes to compute the fluxes every 1 nm in the range [0.4, 0.7] μm. Eventually, the 1 nm fluxes are summed up to yield the PAR.

A numerical validation of the new method was performed in Wandji Nyamsi et al. [30] by comparing the results to the PAR fluxes given by the detailed spectral calculations in libRadtran. Both the relative bias in the absolute value and the root mean square error reached a maximum of 1% for both the direct and global PAR fluxes. It was concluded that the new method performs very well in assessing the global PAR and its two components in these numerical experiments, and that it is computationally much less demanding than the detailed spectral calculations in libRadtran.

## 4. Results of the Validation

The estimates from the new method were validated against 1 min global PPFD measurements for cloudless conditions. For each pair of instantaneous data, the deviation, i.e., estimate minus measurement, was computed. Various statistics were then calculated: the bias (mean of the deviations), the root mean square difference (RMSD), their relative values, respectively rbias and rRMSD, with respect to the mean value of the measurements, and the squared correlation coefficient ($R^2$). Changes in the results with the month and the year were also investigated.

Figure 2 exhibits the 2D histogram, also known as scatter density plot, between the measured and estimated global PPFD at Fort Peck. The points lie mostly along the 1:1 line, and the slope of the fitting line is 1.03. However, there is a tendency to overestimate the greatest PPFD. The estimates and measurements are very well correlated with $R^2$ equal to 0.98 (Table 3). The bias and rbias are small: +11 $\mu$mol m$^{-2}$ s$^{-1}$, and. +1% respectively. The RMSD and rRMSD are also small: 58 $\mu$mol m$^{-2}$ s$^{-1}$, and 5% respectively. These quantities fluctuate slightly from one year to the next, and do not exhibit any trend. These results demonstrate the high accuracy and very good ability of the new method to reproduce the PPFD and its temporal variability.

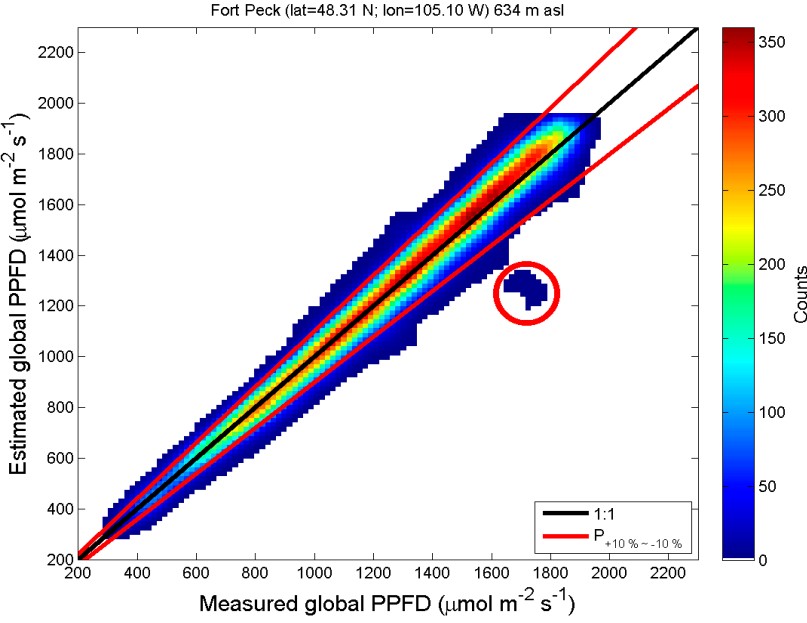

**Figure 2.** 2D histogram of measured photosynthetic photon flux density (PPFD) and estimates at Fort Peck in cloudless atmospheres. The color bar indicates the number of couples in each bin of 20 $\mu$mol m$^{-2}$ s$^{-1}$ in width.

**Table 3.** Statistics of comparison between the measured global PPFD and estimates. N is the number of data pairs.

| Station | N | Mean | Bias | RMSD | rBias (%) | rRMSD (%) | $R^2$ |
|---|---|---|---|---|---|---|---|
| Fort Peck | 186,698 | 1262 | 11 | 58 | 1 | 5 | 0.98 |
| Sioux Falls | 245,355 | 1247 | 1 | 53 | 0 | 4 | 0.98 |
| Penn. State Univ | 120,097 | 1273 | 61 | 82 | 5 | 6 | 0.98 |
| Table Mountain | 260,509 | 1263 | 50 | 69 | 4 | 5 | 0.99 |
| Bondville | 196,871 | 1257 | 36 | 74 | 3 | 6 | 0.97 |
| Desert Rock | 603,727 | 1424 | −12 | 37 | −1 | 3 | 0.99 |
| Goodwin Creek | 230,420 | 1320 | 42 | 70 | 3 | 5 | 0.98 |

The area delimited by two red lines in Figure 2 represents the relative errors within ±10%. One can observe that most of the points fall in that area. A red circle points out a few points exhibiting more than a 20% underestimation. All these points belong to a single day: 02-07-2015, during which, the CAMS and measured AOD at 500 nm were 1.8 and 0.6 in average, respectively. Because of the major contribution of the direct PPFD to the global PPFD in cloudless conditions, and in view of the fact that the direct is strongly dependent of the aerosol load, this overestimation of the AOD by CAMS may explain the observed underestimations.

The influence of inputs on errors in the PPFD was also examined. Figure 3 exhibits the mean, 1st, 2nd, and 3rd quartiles of the ratio (estimate/measurement) and deviation for various classes of $\theta_s$, albedo, TOC, TWV, and AOD at 550 nm. Overall, no evident dependency of errors with the tested variables is found. AOD is an exception, and errors exhibit a tendency to get more negative with an increasing AOD. For both the ratios and the deviations, the sizes of the boxes for a given variable are small, meaning that there is a very limited spread of errors. In addition, these sizes are fairly similar from one interval to another, meaning that there is a weak dependency of the spread of errors with the variable, except for a large AOD.

Figure 4 exhibits 2D histograms at the other stations. The points in the graph are elongated along the 1:1 line, with a vast majority falling within ±10%. At all stations, $R^2$ is always greater than 0.97 (Table 3), meaning that the estimates reproduce well the variability in the PPFD. An overestimation is observed as a whole in Figure 4, and its magnitude depends on the station. At Sioux Falls and Desert Rock, similarly to Fort Peck, one observes a tendency to slightly overestimate the PPFD that is greater than 1500 µmol m$^{-2}$ s$^{-1}$. The tendency is more marked at the other stations for the PPFD that is greater than 1000 µmol m$^{-2}$ s$^{-1}$. The bias ranges between a minimum of −12 µmol m$^{-2}$ s$^{-1}$ (−1% at Desert Rock) and a maximum of 61 µmol m$^{-2}$ s$^{-1}$ (5% at Penn. State. Univ.).

Penn. State. Univ exhibits the greatest bias. The ground there is covered by crops and grass most of the time. In such cases, the mean ratio between the PAR albedo and the broadband one should be close to 0.2–0.3 (Bosch et al. [22]) instead of 0.47 as used here. Using a smaller ratio would yield a smaller PAR albedo, a smaller contribution of the flux reflected by the ground to the diffuse PAR, a smaller global PAR, and eventually a smaller bias.

No clear dependencies were found between the results and month or year. The changes in errors with the $\theta_s$, TOC, albedo and AOD were also examined (not shown). The maxima of ratios or deviations are less at the six stations than at Fort Peck because of the large underestimation in the AOD on 02-07-2015. Otherwise, the results are similar to those at Fort Peck (Figure 3).

One may expect a dependency of the bias and RMSD with $\theta_s$. The bias and RMSD and their relative values vary from positive values to negative values as a function of $\theta_s$, at each station (not shown). Nevertheless, they are kept within ±6% in relative values, which demonstrates a limited influence of $\theta_s$.

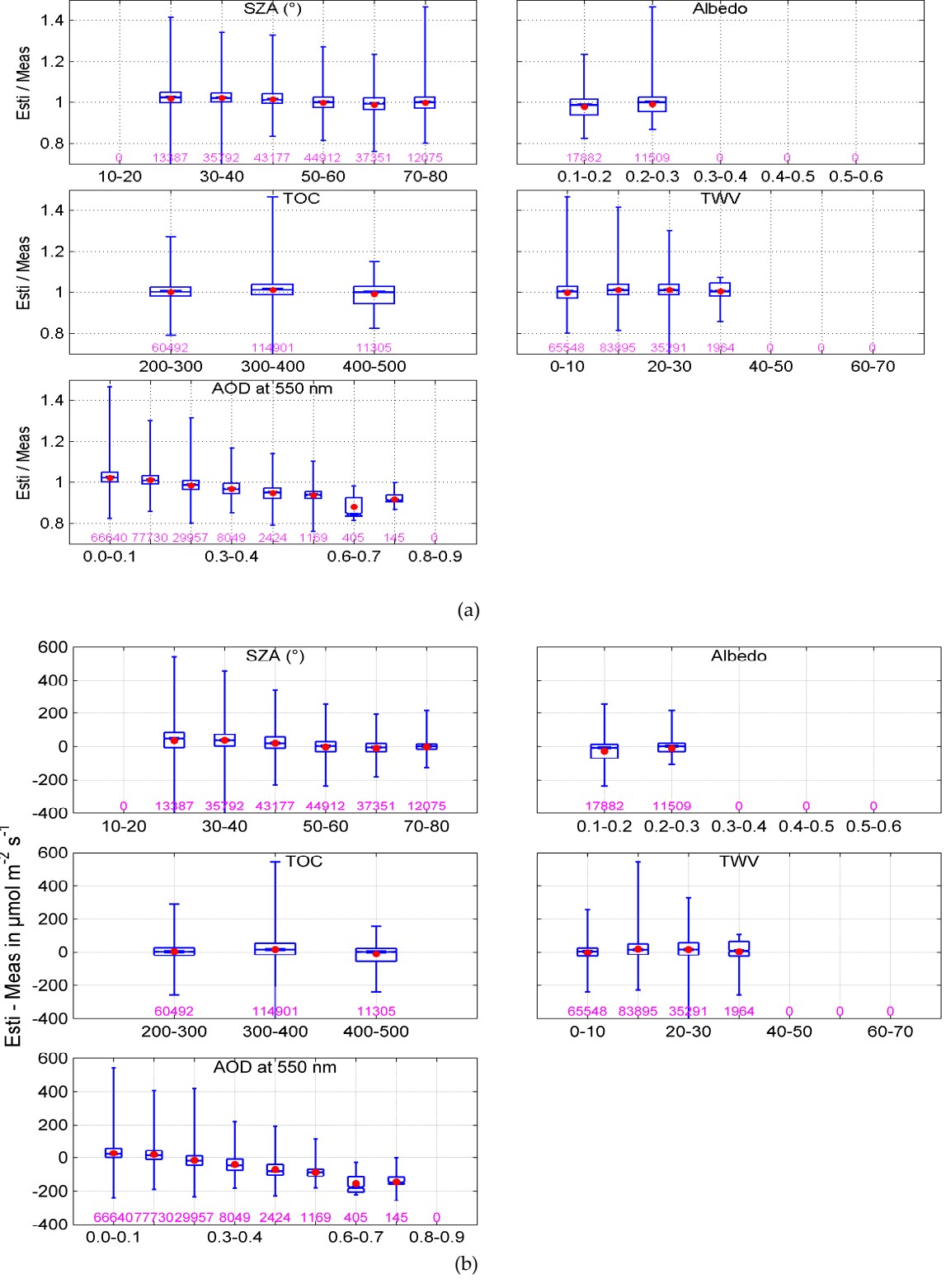

**Figure 3.** Change in ratio (estimate/measurement) (**a**) and deviation (estimate-measurement) (**b**) as a function of the solar zenith angle (SZA), albedo, total ozone column (TOC), total water vapor (TWV) and aerosol optical depth (AOD) at Fort Peck. Red dots indicate the mean. The limits of the boxes give the 1st, 2nd, and 3rd quartiles, while the extreme whiskers are the minimum and maximum. The numbers of pairs are given in pink.



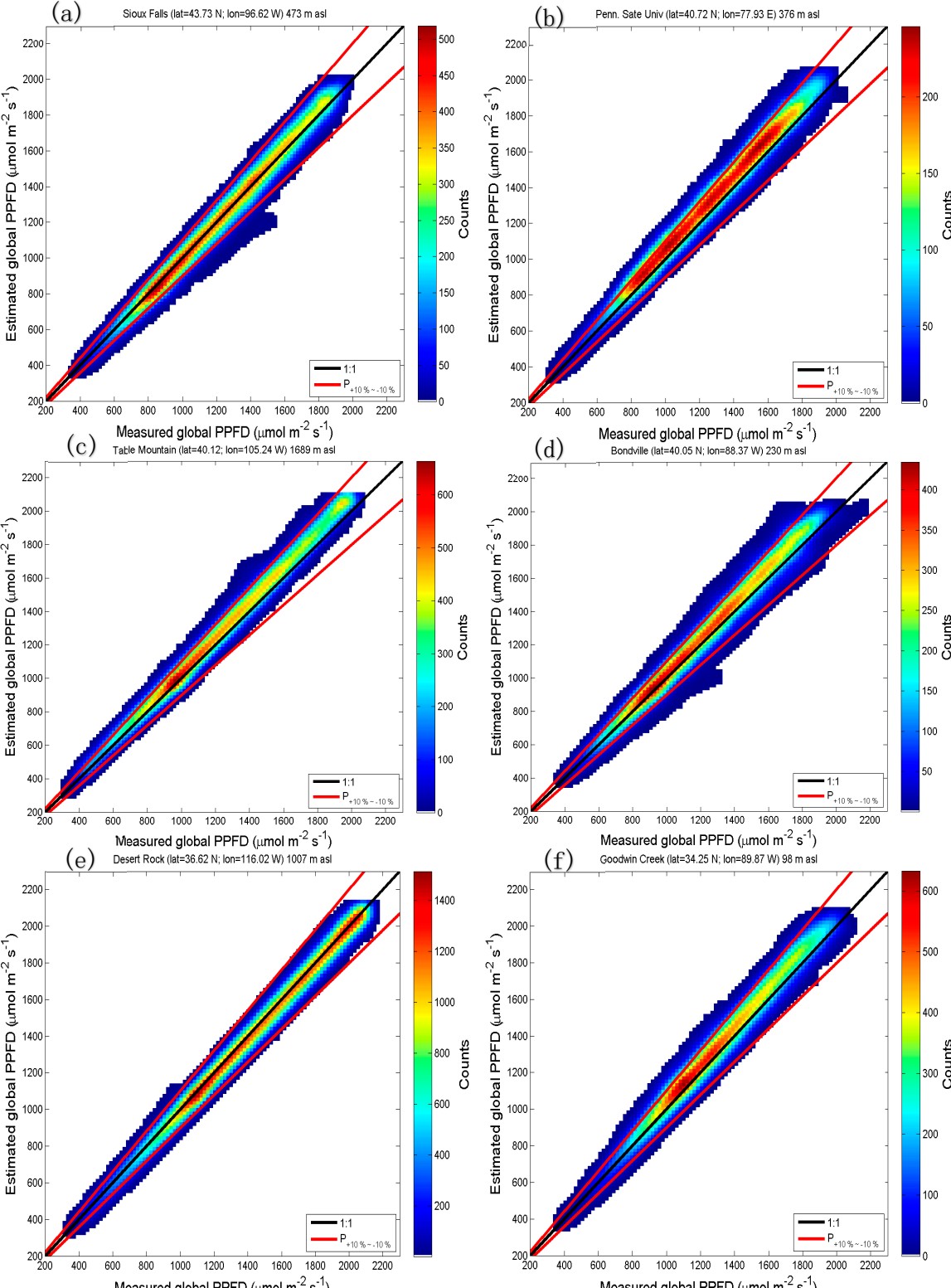

**Figure 4.** 2D histogram of the measured PPFD and estimates in cloudless atmospheres at each station except Fort Peck: (**a**) Sioux Falls; (**b**) Penn. State Univ.; (**c**) Table Mountain; (**d**) Bondville; (**e**) Desert Rock; (**f**) Goodwin Creek. The color bar indicates the number of couples in each bin of 20 µmol m$^{-2}$ s$^{-1}$ in width.

## 5. Discussion

When the PAR estimates from the technique were compared against the values obtained by the detailed spectral calculations of libRadtran, Wandji Nyamsi et al. [30] found no systematic bias as a function of any inputs. In addition, there was no systematic bias or trend as a function of the PAR intensity. Therefore, and since the SG2 algorithm giving $\theta_s$ is very accurate, the overestimation of the greatest PAR fluxes may be related to the errors in the CAMS products serving as inputs to the method as well as the assumption on the PAR albedo.

Comparisons between the CAMS products and their counter-parts measured at the SURFRAD sites or at the closest AErosol RObotic NETwork (AERONET) sites were carried out. The results show that there is no clear dependency of the errors with the uncertainty of the CAMS TWV and AOD. Therefore, the overestimation may be related to the assumed PAR-albedo computed with a constant of 0.47, which realistically should depend on the type of surface. Unfortunately, there were no PAR-albedo measurements available at the SURFRAD stations to check this hypothesis.

As stated in the introduction, only Su et al. [20] results can be directly compared to our results, as the inputs are also satellite-based measurements: CERES for the Su et al. [20] method and CAMS for our method. The validations by Su et al. [20] were performed over the period from March 2000 to June 2005 at the same seven SURFRAD sites. In our case, they were performed from January 2010 to December 2016. A rigorous assessment of the performance between methods needs to be done over a similar time period. Unfortunately, our method is only applicable after 2003, when the CAMS products are available. Nevertheless, the relative performances can be compared. The relative biases for the Su et al. [20] method (respectively our method) were 0% (+1%) at Fort Peck, +3% (0%) at Sioux Falls, +5% (+5%) at Penn State, +2% (+4%) at Table Mountain, +7% (+3%) at Bondville, +2% (−1%) at Desert Rock, and +4% (+3%) at Goodwin Creek. One may conclude that our method shows a similar or better performance than the Su et al. [20] method.

## 6. Conclusions

A new method for assessing PAR fluxes in cloudless atmospheres has been presented and evaluated here. It is a first step towards an entire ready-to-use tool for assessing the PAR fluxes in all-sky conditions on a routine basis, that is similar to what is currently done within the CAMS Radiation Service ([38]), thus providing users with an easy access to PAR fluxes on a global scale. The atmospheric inputs are the total column contents in ozone and water vapor, and the aerosol properties provided by the CAMS and the ground reflective properties are excerpted from the MODIS-derived data sets of BRDF parameters from Blanc et al. [31].

The method has been validated by comparing its outputs to measured 1 min global PAR at seven stations located within various climates in the USA. At all stations, the squared correlation coefficient exceeds 0.97, demonstrating that the vast majority of the temporal variability is well reproduced by the proposed method. The relative bias varies from −1% to +5%. The relative RMSD is very close to the relative bias, indicating a very small standard deviation of errors. The relative values of the standard deviation are under 5%. If the statistical distribution of the errors is gaussian, then the uncertainty (percentile 95) is under 10%, i.e., close to the uncertainty of the measurements (8%). In addition, our results were compared to state-of-the-art clear-sky methods of the PAR estimation. They show that our method can offer similar and even better performances than the other methods. The results of the validation demonstrate the high quality of the PAR estimates from our method.

This method is capable of estimating the global PAR as well as its direct and diffuse components, though the validation was carried out only on global PAR fluxes. The PAR fluxes for each KB could be obtained quite rapidly by taking advantage of the pre-computed abaci made for the clear-sky McClear model, which is $10^5$ times faster than libRadtran [29].

Because the proposed method offers accurate estimates in cloudless atmospheres, one benefit is that any modelling of the attenuation due to clouds may be combined with our method to provide all–sky PAR estimates. Examples of such a combination are the clear-sky index and cloud modification

factor, as discussed by Oumbe et al. [19] or Huang et al. [39] for total irradiance, or the UV range discussed by Calbo et al. [40], den Outer et al. [41] or Krotkov et al. [42]. The approach suggested by Wandji Nyamsi et al. [37] for the surface albedo in UV could be another means of improving PAR estimates in cloudless and all–sky conditions.

**Author Contributions:** W.W.N. conceived the presented method, which was designed with help from all co-authors. W.W.N., P.B. and L.W. implemented the method. W.W.N., P.B., J.A.A., A.A. and L.W. participated in writing and editing the manuscript, as well as investigating and interpreting the results.

**Funding:** This research received no external funding.

**Acknowledgments:** The authors thank the NOAA ESRL Global Monitoring Division, Boulder, Colorado, USA (http://esrl.noaa.gov/gmd/, last access: 14-04-2019) for offering free access to SURFRAD data.

**Conflicts of Interest:** The authors declare no conflict of interest.

**Data Availability:** PAR measurements at each station were provided by the SURFRAD network established in 1993 through the support of the NOAA Office of Global Programs. Measurements used here are freely available and were downloaded from ftp://aftp.cmdl.noaa.gov/data/radiation/surfrad/, last access: 14-04-2019. Products from CAMS are freely available after registration and were downloaded from: http://atmosphere.copernicus.eu/, last access: 14-04-2019. The McClear products are freely available after registration and were downloaded from: http://www.soda-pro.com, last access: 04-14-2019. The BRDF maps by Blanc et al. [31] may be downloaded from: http://www.oie.mines-paristech.fr/Valorisation/Outils/AlbedoSol/, last access: 14-04-2019.

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
