# Peer review of "A New Clear-Sky Method for Assessing Photosynthetically Active Radiation at the Surface Level"

_atmosphere, doi:10.3390/atmos10040219_

Round 1

Reviewer 1 Report

This is an interesting research paper validating a clear–sky method to estimate the photosynthetically active radiation at surface level in cloudless atmosphere. 

 Its outcomes were successfully compared  to instantaneous global photosynthetic photon flux density (PPFD) measurements acquired at seven  experiment sites of the Surface Radiation network (SURFRAD) in the US. 

Author Response

Dear Reviewer #1

We thank you for your positive comments on the manuscript.

With best regards,

William Wandji Nyamsi

Reviewer 2 Report

Please include pronouns where warranted (e.g., in the Abstract, ...at surface level in a cloudless atmosphere ...). Please include commas where warranted (e.g., in the Abstract, ...accurate approximation, adopted in several...as the k-distribution method, and the ...). Always use a comma after 'et al.' (e.g., in the Abstract, ...Kato et al., (1999), that gives...). In the Abstract, the statement should be, "acquired at seven experimental sites ...". Last sentence in the Abstract should read, "...in cloudless atmospheres at a high...".

In the Introduction, provide a reference noting the role that PAR plays in plant photosynthesis. PAR is the incident power per unit surface area. Second paragraph of Introduction (first sentence) should read "...domains demand high quality estimates...". Third paragraph of Introduction (second sentence) should read "Because of instrument costs, ...". Third paragraph of Introduction (third sentence) should read "...proxies for PAR...".  Please look out for the above type of issues in the manuscript.

In the third paragraph of the Introduction, add a reference for quantum sensors and/or their use in similar studies. Third paragraph of Introduction (sixth sentence) should read "...clear benefit of using...". Always use a period after 'e.g.' or 'i.e.'.  Usage should be , "...cloudless  atmospheres..." or, "...a cloudless atmosphere...". Always use the following, "experimental sites".  Usage should be, "...negative bias of less than...".  Usage should be, "...ground-based measurements as inputs is that their accuracy is better, which translates...". Always use commas before conjunctions (e.g., and, or, but, etc.). Usage should be, "...provided an accurate description...". Define the acronym (KNMI - Koninklijk Nederlands Meteorologisch Instituut - Royal Netherlands Meteorological Institute).  Missing period, "produce broadband irradiance. In this scheme, ...".  Usage should be , "...we advance in a step-by-step approach, and...".  Usage should be "...assess the performance of the clear-sky algorithm, to be better able to understand later, the performance of the modelling...".  Usage should be "...column contents of ...". Define the acronym BRDF - Bi-directional Reflectance Distribution Function at its first use. Usage should be, "...SURFRAD stations used, while Table 1 reports...". Ensure that all weblinks are active (blue) in the manuscript and ensure that you consistently provide the date when each weblink was last accessed (e.g., " - Accessed: 02 April 2019."). Use US date formats, not European formats (e.g., use: 01-01-2010 to 12-31-2016). Ensure that text font is consistent through the manuscript (see weblink under Figure 1, and the bottom half of page 5). When referring to numerical values, use 'approximately' instead of 'about'. Usage: use "solar zenith angle". Usage, "The resampling technique has been presented..." or, "The resampling techniques have been presented ...". Use a comma to denote numerical values with 4 or more digits, e.g., 1,000, 60,000, etc.). Usage, "...atmospheric condition parameters in cloudless atmospheres has been built...". Usage ".. converted into transmissives...". Usage, "... dependency on the extraterrestrial...". Add spaces for paragraph starting under Table 2. Ensure that words requiring pluralization are pluralized. Usage: use, "in assessing" instead of "to assess" in the sentence before Section 4. Add spaces for paragraph starting under Table 3. Add spaces to separate titles for last two graphics in Figure 4 on page 11. Ensure there is a space after each semicolon and colon. Add periods after the sentences at the bottom of page 13.

Add spaces between the references in the References section, ensure that the reference format is correct, and that the references are single-spaced.. 

Author Response

Dear Reviewer #2,

First of all, we thank Reviewer #2 for the constructive comments and remarks on this article. The authors believe that they have understood the concerns of the reviewer. The remarks have been taken into account for revising a part of the text following recommendations of the reviewer.

Point 1: Please include pronouns where warranted (e.g., in the Abstract ...at surface level in a cloudless atmosphere ...). Please include commas where warranted (e.g., in the Abstract ...accurate approximation, adopted in several...as the k-distribution method, and the ...). Always use a comma after 'et al.' (e.g., in the Abstract ...Kato et al., (1999), that gives...). In the Abstract, the statement should be, "acquired at seven experimental sites ...". Last sentence in the Abstract should read, "...in cloudless atmospheres at a high...".

Thank you for these remarks. Done as requested. In addition, we used comma after every 'et al.' (i.e. Smith et al., 2007 for instance) for regular citations. However, Kato et al. (1999) is referring to the method/approach and thus is not a regular citation. This notation without comma was several times accepted in our similar peer reviewed papers. Please, referred to:

1.     Wandji Nyamsi, W.; Espinar, B.; Blanc, P.; Wald, L. How close to detailed spectral calculations is the k‑distribution method and correlated‑k approximation of Kato et al. (1999) in each spectral interval?. Meteorol. Z. 2014, 23, 547-556. doi: 10.1127/metz/2014/0607.

2.     Wandji Nyamsi, W.; Espinar, B.; Blanc, P.; Wald, L. Estimating the photosynthetically active radiation under clear skies by means of a new approach. Adv. Sci. Res. 2015, 12, 5-10. doi:10.5194/asr-12-5-2015.

3.     Wandji Nyamsi, W.; Pitkänen, M.; Aoun, Y.; Blanc, P.; Heikkilä, A.; Lakkala, K.; Bernhard, G.; Koskela, T.; Lindfors, A.; Arola, A.; Wald, L. A new method for estimating UV fluxes at ground level in cloud-free conditions. Atmos. Meas. Tech. 2017, 10, 4965-4978. doi: 10.5194/amt-10-4965-2017

So we did not make changes to the Kato et al. (1999) now. However, depending on the specific notation, in this case, used in Atmosphere and requested by the editorial office, we are certainly glad to modify the text accordingly.

Point 2: In the Introduction, provide a reference noting the role that PAR plays in plant photosynthesis. PAR is the incident power per unit surface area. Second paragraph of Introduction (first sentence) should read "...domains demand high quality estimates...". Third paragraph of Introduction (second sentence) should read "Because of instrument costs, ...". Third paragraph of Introduction (third sentence) should read "...proxies for PAR...".  Please look out for the above type of issues in the manuscript.

Thank you for this remark. We have changed the text accordingly.

Point 3: In the third paragraph of the Introduction, add a reference for quantum sensors and/or their use in similar studies. Third paragraph of Introduction (sixth sentence) should read "...clear benefit of using...". Always use a period after 'e.g.' or 'i.e.'.  Usage should be , "...cloudless  atmospheres..." or, "...a cloudless atmosphere...". Always use the following, "experimental sites".  Usage should be, "...negative bias of less than...".  Usage should be, "...ground-based measurements as inputs is that their accuracy is better, which translates...". Always use commas before conjunctions (e.g., and, or, but, etc.). Usage should be, "...provided an accurate description...". Define the acronym (KNMI - Koninklijk Nederlands Meteorologisch Instituut - Royal Netherlands Meteorological Institute).  Missing period, "produce broadband irradiance. In this scheme, ...".  Usage should be , "...we advance in a step-by-step approach, and...".  Usage should be "...assess the performance of the clear-sky algorithm, to be better able to understand later, the performance of the modelling...".  Usage should be "...column contents of ...". Define the acronym BRDF - Bi-directional Reflectance Distribution Function at its first use. Usage should be, "...SURFRAD stations used, while Table 1 reports...". Ensure that all weblinks are active (blue) in the manuscript and ensure that you consistently provide the date when each weblink was last accessed (e.g., " - Accessed: 02 April 2019."). Use US date formats, not European formats (e.g., use: 01-01-2010 to 12-31-2016). Ensure that text font is consistent through the manuscript (see weblink under Figure 1, and the bottom half of page 5). When referring to numerical values, use 'approximately' instead of 'about'. Usage: use "solar zenith angle". Usage, "The resampling technique has been presented..." or, "The resampling techniques have been presented ...". Use a comma to denote numerical values with 4 or more digits, e.g., 1,000, 60,000, etc.). Usage, "...atmospheric condition parameters in cloudless atmospheres has been built...". Usage ".. converted into transmissives...". Usage, "... dependency on the extraterrestrial...". Add spaces for paragraph starting under Table 2. Ensure that words requiring pluralization are pluralized. Usage: use, "in assessing" instead of "to assess" in the sentence before Section 4. Add spaces for paragraph starting under Table 3. Add spaces to separate titles for last two graphics in Figure 4 on page 11. Ensure there is a space after each semicolon and colon. Add periods after the sentences at the bottom of page 13.

Thank you for these suggestions and remarks. We have re-written the relevant part of the text accordingly. We have added a reference for quantum sensors as follows: https://www.licor.com/documents/3bjwy50xsb49jqof0wz4

Point 4: Add spaces between the references in the References section, ensure that the reference format is correct, and that the references are single-spaced.

Thank you for this remark. Done as requested.

With best regards,

William Wandji Nyamsi
